# Preserving Representation in Continual Learning via Feature-Preserving Fine-Tuning

## Abstract

In real-world applications, deep learning models must continually adapt to sequentially arriving tasks without access to previous data. Although pre-trained foundation models show generalisation and zero-shot abilities, fine tuning them in a continual learning setting often leads to representation degradation. In this study, we firstly systematically evaluate several recent feature-preserving fine-tuning methods (L2-SP, FTP, WiseFT and ImpReg) in continual learning scenario using a large scale pre-trained foundation model. We further explore the effectiveness of full fine-tuning (FullFT) versus parameter-efficient fine tuning (PEFT) and propose a novel two-stage fine-tuning strategy, PEFT+Cons, designed to balance stability and plasticity by combining PEFT with task-specific knowledge consolidation. Extensive experiments on the CIFAR-100 and ImageNet-R benchmark datasets demonstrate that our proposed PEFT+Cons approach effectively prevents representation forgetting while enhancing task-specific knowledge retention.

## 1 Introduction

Traditional deep learning models are typically trained in static, offline settings where the entire training dataset is assumed to be available in advance. This assumption diverges from real-world scenarios where data arrives in a non-stationary, sequential, or evolving manner, limiting their applications in dynamic environments. *Continual learning* (Parisi et al., 2019) addresses this limitation by enabling models to learn from a stream of tasks over time without revisiting the entire dataset. However, it faces the challenge of catastrophic forgetting. (Nguyen et al., 2019; De Lange et al., 2021), where models tend to lose previously acquired knowledge while adapting to new data.

Deep learning models operate across three conceptual spaces: input, representation and output. As high-level abstractions are difficult to infer directly from raw input data, models learn intermediate representations that capture semantically rich, task-relevant features for downstream predictions. Traditional continual learning research has focused on maintaining output performance (e.g. classification accuracy across tasks), yet preserving the quality of learned representations is equally critical, especially with the advances of pre-trained foundation models in recent years. Models such as CLIP (Radford et al., 2021), DINO (Caron et al., 2021), and MAE (He et al., 2022) are trained on massive, diverse datasets and aim to produce generalisable features that transfer effectively to downstream tasks. While they demonstrate strong zero-shot generalisation (Thengane et al., 2022), fine-tuning is required to optimise performance for specific domains. Fine-tuning strategies can be divided into two categories: **Full Fine-Tuning (FullFT)**, where all parameters are updated (Thengane et al., 2022; Xuhong et al., 2018; Tian et al., 2023; Wortsman et al., 2022; Li et al., 2024), and **Parameter-Efficient Fine-Tuning (PEFT)**, where only a small task-specific subset of parameters is adapted (e.g. adapters, prompt and final layers) while the backbone is frozen (Wang et al., 2022a;b; Smith et al., 2023b; Huang et al., 2025; Smith et al., 2023a).

Although FullFT has been found to offer greater flexibility and often achieve higher performance, it can degrade the generalisation ability of pre-trained representations (Kumar et al., 2022), especially in continual learning where sequential adaptation is required. This representation forgetting (Davari et al., 2022) is highly undesirable, as the model may progressively lose its pretrained generalisation capacity learned during pre-training and become unusable for future tasks. To address this, recent work has proposed feature-preserving methods that balance plasticity (adaptation) and stability (retention) by constraining updates to remain close to the original pretrained weights, typically via

parameter-space regularisation. By limiting the degree to which the pre-trained weights are altered, the model keeps the generalisable representation that were acquired during pre-training.

Motivated by this, we study recent feature-preserving approaches that aim to retain the robustness and generalisation of pre-trained models while incorporating new knowledge. Namely, we examine four methods: **L2-SP** (Xuhong et al., 2018), **FTP** (Tian et al., 2023), **WiseFT** (Wortsman et al., 2022) and **ImpReg** (Li et al., 2024). Although they have shown promise in offline, single-task fine-tuning, their effectiveness in continual learning scenarios remains largely unexplored. The sequential and non-stationary nature of continual learning introduces additional challenges, such as representation forgetting, task interference, and unseen-task generalisation, which are not present in offline fine-tuning. As a result, it remains unclear whether feature-preserving techniques can adequately maintain the representation quality across multiple tasks in continual learning setting.

In this paper, we investigate feature-preserving methods in continual learning using a CLIP ResNet-50 model pre-trained on ImageNet-1K. We evaluate their effectiveness under class-incremental learning for image classification across two benchmarks, CIFAR-100 (Krizhevsky & Hinton, 2009) and ImageNet-R (Hendrycks et al., 2021), where each dataset is split into 10 disjoint tasks. To assess robustness and generalisability, we employ representation-level metrics such as linear probe accuracy, representational forgetting, and unseen-task generalisation. Our contributions are as follows:

1. We evaluate four recent feature-preserving fine-tuning methods: **L2-SP, FTP, WiseFT and ImpReg**, in a continual learning setting with pre-trained foundational models, highlighting their strengths and limitations when applied to pre-trained models.

2. We investigate how these methods affect representation robustness under two fine-tuning strategies: **FullFT** and **PEFT**, focusing on the role of the attention pooling block. We assess representational forgetting, linear probe performance and unseen tasks generalisation.

3. We propose **PEFT+Cons**, a novel two-stage fine tuning strategy that combines PEFT with task-specific knowledge consolidation, and provide insights into its effectiveness in mitigating representational forgetting.

## 2 RELATED WORK

Continual learning refers to methods aiming to address the challenge of catastrophic forgetting in machine learning models, enabling them to learn tasks sequentially. Existing approaches are often grouped into three categories: architectural expansion, rehearsal-based, and rehearsal-free.

**Architectural expansion methods** (Rusu et al., 2016; Yoon et al., 2018; Mallya & Lazebnik, 2018) dynamically expand the model's architecture as new tasks arrive. While this can be effective in avoiding interference between tasks, the drawback is that the model size and computational cost grows as the number of tasks increases. We focus on approaches that update the existing model parameters without architectural expansion as they offer more scalability in constrained settings.

**Rehearsal-based methods** (Rebuffi et al., 2017; Lopez-Paz & Ranzato, 2017; Hayes et al., 2019; Harun et al., 2023; Rolnick et al., 2019) address forgetting by storing and replaying a subset of past data, which has proven highly effective in continual learning settings. Several works have explored different heuristics for selecting and maintaining replay buffers in a resource-efficient manner or even use generative models to synthesize past examples (Shin et al., 2017). While highly effective, these approaches raise storage and privacy concerns (e.g. private data that cannot be stored long term) (Wang et al., 2024), especially in the context of foundation models such as CLIP (Radford et al., 2021), where the original training data is closed-source and unavailable for replay. This motivates us to explore rehearsal-free approaches that do not depend on sample storage.

**Rehearsal-free methods** avoid explicit storage by relying on regularisation (Kirkpatrick et al., 2017; Aljundi et al., 2018; Zenke et al., 2017) or knowledge distillation (Li & Hoiem, 2017) to preserve past knowledge. Their appeal lies in eliminating the need for replay buffers, making them suitable for privacy-sensitive or memory-constrained settings. However, they often underperformed compared to rehearsal-based methods in complex or long-tailed scenarios (Smith et al., 2023a).

To address this gap, recent work has begun utilising *pre-trained foundational models within rehearsal-free frameworks*. Several works leverage pre-trained models by freezing their parame-

ters during continual learning (Prabhu et al., 2023; Ostapenko et al., 2022), preventing forgetting by keeping representations fixed. This approach is resistant to forgetting as the frozen parameters remain unchanged. However, this assumes that the pre-trained representations generalise well across all tasks in continual learning. This assumption often doesn't hold in practical applications. As such, there is a need for fine-tuned approaches that can continually learn new knowledge as well as retaining the pre-trained knowledge. Other works have explored prompt tuning techniques for continual learning (Wang et al., 2022a;b; Smith et al., 2023b; Huang et al., 2025).While these methods have shown to be highly effective in continual learning settings, they keep the pre-trained model frozen and only tune the learnable parameters of the added prompts. In this paper, we investigate whether updating the existing parameters of a pre-trained model can better balance retention of prior knowledge while adapting to new tasks.

## 3 METHODOLOGY

Our overall goal is to develop a deeper understanding of how feature-preserving methods behave under different fine-tuning strategies in continual learning settings. In this section, we formally introduce the problem and the performance measures, and describe the four feature preserving methods that we compare (L2-SP, FTP, WiseFT and ImpReg) under two different fine-tuning strategies (FullFT and PEFT). The results are presented and discussed in Sec. 4.

### 3.1 CONTINUAL LEARNING AND CLASS-INCREMENTAL LEARNING

In continual learning, a model encounters a sequence of tasks $\mathcal{T} = \{\mathcal{T}_1, \mathcal{T}_2, \ldots, \mathcal{T}_N\}$, where each task $\mathcal{T}_i$ is associated with a training dataset $\mathcal{D}_i^{\text{train}} = \{(\boldsymbol{x}_j, y_j)\}_{j=1}^{n_i}$, as well as a respective test dataset $\mathcal{D}_i^{\text{test}}$. A key constraint in continual learning is that once a task is completed, its data is no longer accessible, prohibiting joint training across tasks.

We focus on *class-incremental learning*, where each task introduces a disjoint subset of classes from the overall class set $\mathcal{C} = \bigcup_{i=1}^{N} \mathcal{C}_i$, with $\mathcal{C}_i \cap \mathcal{C}_j = \emptyset$ for $i \neq j$. In our setting, we use a single shared classification head throughout sequential training, which is a commonly used in class-incremental learning settings. The shared head is incrementally expanded as new classes are introduced.

Following prior work, we apply the *"labels trick"* (Zeno et al., 2021) during training to minimise task bias, ensuring fairness across early and later tasks without violating the class-incremental protocol at test time, which has shown to significantly improve the task accuracy. Note that this method is applied during training, using access to the known task identity. At test time, the task identity becomes unavailable, thus not breaking the class-incremental learning protocol.

### 3.2 EVALUATING REPRESENTATION

To assess how well the model maintains its representation quality throughout continual learning, we decompose it into an encoder $f_{\boldsymbol{\theta}} : \mathcal{X} \to \mathcal{Z}$ and a classifier $g_{\boldsymbol{\phi}} : \mathcal{Z} \to \mathcal{Y}$. Rather than focusing only on classifier accuracy, we examine changes in the representation space by training a linear probe (LP) classifier at each task. We propose the following metrics:

- **Final Task Accuracy (FTA):** the average LP accuracy across all tasks after the final update, providing a holistic measure of how well the final encoder retains information across the entire sequence:

$$\text{FTA} = \frac{1}{T} \sum_{i=0}^{T-1} A_{T-1,i} \tag{1}$$

- **Representational Forgetting (RF):** measures the drop in LP accuracy for each task from when it was first learned to the end of training. Formally, for each task $\mathcal{T}_i$, forgetting is defined as $A_{i,i} - A_{T-1,i}$, and the total representational forgetting is:

$$\text{RF} = \sum_{i=0}^{T-1} \left( A_{i,i} - A_{T-1,i} \right) \tag{2}$$

RF quantifies the degradation of representations as the model sequentially learns new tasks.

- **Unseen Tasks Accuracy (AvgUTA):** measures how well the current model produces representations that generalise to tasks it has not yet been trained on:

$$\text{UTA}(b) = \frac{1}{|U_b|} \sum_{i \in U_b} A_{b,i},$$

(3)

where $U_b = \{\mathcal{T}_i \mid i > b\}$ denotes the set of unseen tasks, aggregated over all frozen encoders at each task boundary,

$$\text{AvgUTA} = \frac{1}{T-1} \sum_{b=0}^{T-2} \text{UTA}(b)$$

(4)

AvgUTA assesses how well the encoder generalises to future tasks it has not yet been trained on, thereby capturing the generalisation ability of the current representation.

- **Global LP Accuracy (FinalGLP):** measures the LP accuracy when it is trained on the entire dataset at frozen encoder $f_{\boldsymbol{\theta}_b}$. Specifically, we train a linear probe $h_{\boldsymbol{\delta}_b}(f_{\boldsymbol{\theta}_b}(\boldsymbol{x})) : \mathcal{Z} \to \mathcal{Y}$, which produces the test accuracy $\text{GLP}_b$ at task boundary $b$.

$$\text{FinalGLP} = \text{GLP}_{T-1}$$

(5)

FinalGLP reflects the overall linear separability of the representation space learned by the end of the entire task sequence. This is crucial for understanding how well the learned representations have retained information once all task boundaries are removed.

### 3.3 FEATURE-PRESERVING METHODS

We study four notable methods designed to prevent representation degradation during fine-tuning:

- **L2-SP** (Xuhong et al., 2018): introduces an $L_2$ penalty that anchors the fine-tuned weights to the pre-trained parameters. Unlike standard weight decay, L2-SP directly regularises distance from the initial model.

- **FTP** (Tian et al., 2023): a projection-based method that first applies unconstrained gradient updates and then projects the weights back relative to the pre-trained parameters.

- **WiseFT** (Wortsman et al., 2022): interpolates between pre-trained and fine-tuned weights.

- **ImpReg** (Li et al., 2024): regularises updates based on neuron importance estimated from the gradients. Parameters associated with critical neurons are constrained more heavily.

In our continual learning setting, we adapt these methods to preserve the representation quality across task sequence by treating the model parameters from the previous task $\boldsymbol{\theta}_{t-1}$ as a new reference point, replacing the original pre-trained weights $\boldsymbol{\theta}_0$ at the beginning of each new task $\mathcal{T}_i$.

### 3.4 FINE-TUNING STRATEGIES

We evaluate feature-preserving methods under two different fine-tuning strategies: *full fine-tuning* and *parameter-efficient fine tuning*. These strategies are summarized in Fig.1 and detailed below:

- **Full Fine-Tuning (FullFT):** Under this strategy, all parameters of the encoder $f_\theta$ are updated during training. The model is optimised using the standard training objective, plus the additional feature-preserving constraints.

- **Parameter-Efficient Fine-Tuning (PEFT):** Under this strategy, only the attention pooling block of the encoder $f_\theta$ is updated, while the rest of the network is frozen. The feature-preserving methods are applied only to the parameters being updated.

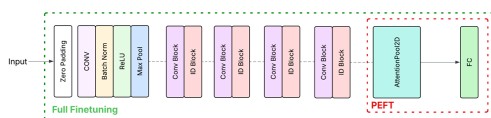 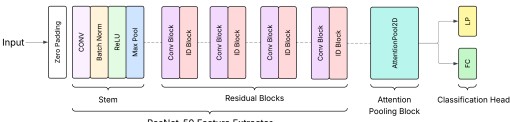

Figure 1: *Fine-tuning strategies applied to the CLIP-ResNet-50*. FullFT (green) updates all layers of the encoder, while PEFT (red) restricts updates to only the attention pooling block, leaving the rest of the encoder frozen.

Figure 2: *CLIP-ResNet-50 architecture*. The stem and residual blocks act as the *Feature Extractor*, the AttentionPool2D as the *Attention Pooling Block* and the final fully connected layer as the *Classification Head*. LP uses the pooling block output to assess representation quality.

## 4  COMPARISON OF FEATURE-PRESERVING METHODS

**Experiment Setup.** We evaluate four feature-preserving methods: L2-SP, FTP, WiSE-FT and ImpReg, on two class-incremental datasets: SplitCIFAR-100 and SplitImageNet-R, each divided into 10 sequential tasks with disjoint class subsets. For each task, the model is trained on the task's training split and evaluated on its test split to measure representation quality.

Our encoder is CLIP-ResNet-50 (Fig. 2), pre-trained on ImageNet-1K. A shared classification head is incrementally expanded as new classes appear. The model is trained using the cross-entropy loss with Adam (learning rate $1 \times 10^{-5}$), batch size 128, and 10 epochs for SplitCIFAR-100 and 20 for ImageNet-R.Default hyperparameters from the original implementations are used for all methods.

At each task boundary, we evaluate representation quality with LP metrics capturing forgetting, generalisation, and degradation. LPs are trained with cross-entropy and Adam for 100 epochs, selecting the top-1 model. All experiments run on a single NVIDIA A100 GPU.

**Datasets.** We use two benchmark datasets: CIFAR-100 (Krizhevsky & Hinton, 2009): 60K images across 100 classes (50K train/10K test) and ImageNet-R (Hendrycks et al., 2021): 100K images across 200 ImageNet-1K classes, featuring non-natural renditions (e.g. art, sketches). For continual learning, each dataset is divided into 10 sequential tasks of disjoint class subsets (10 classes per task for CIFAR-100, 20 per task for ImageNet-R), referred to as SplitCIFAR-100 and SplitImageNet-R.

**Baseline.** To compare the results, we train a baseline model (Naïve fine-tuning) where all model parameters are updated without any constraints.

### 4.1  RESULTS FOR FEATURE-PRESERVING METHODS UNDER FULLFT STRATEGY

To understand the effect of feature-preserving methods in continual learning, we first examine their performance under the FullFT strategy (Table 1 FullFT, Fig. 3). Naïve fine-tuning leads to severe representational forgetting (high RF) and rapid degradation in representation quality. When applying feature-preserving methods,representational forgetting is reduced, and unseen-tasks generalisation (AvgUTA, Table 1) increases relative to Naïve fine-tuning. Among them, WiSE-FT and ImpReg consistently deliver the best results across both datasets, achieving higher FTA, lower RF, and stronger overall representation quality (FinalGLP). FTP shows moderate improvements, while L2-SP provides only marginal gains and in some cases underperforms relative to naïve fine-tuning. However, the issue of representation degradation persisted across all methods, suggesting that FullFT even when constrained remains vulnerable to representational drift over time.

To gain deeper insights, we examined intermediate results across the continual fine-tuning process. In Fig. 4, we observed a general trend that the model suffers catastrophic representational forgetting within a single task once it is trained, and this persists even under stronger feature-preserving methods. While certain methods (Wise-FT, ImpReg) show partial improvements at later stages, these gains are insufficient to offset the severe forgetting that continues to drive overall degradation. These findings raise a question: can representational forgetting be effectively prevented throughout continual learning? Motivated by this, we next investigate fine-tuning under PEFT strategy.

Table 1: Results on SplitCIFAR-100 and SplitImageNet-R for different feature-preserving methods under **FullFT** and **PEFT** strategy.

| Method | SplitCIFAR-100 | | | | SplitImageNet-R | | | |
|---|---|---|---|---|---|---|---|---|
| | FTA | RF | AvgUTA | FinalGLP | FTA | RF | AvgUTA | FinalGLP |
| **FullFT** | | | | | | | | |
| Naïve | 77.6 | 154.3 | 78.4 | 40.6 | 51.5 | 338.3 | 46.2 | 32.9 |
| L2-SP | 76.8 | 159.5 | 80.0 | 38.0 | 48.4 | 440.1 | 45.6 | 30.6 |
| FTP | 80.0 | 135.3 | 81.9 | 44.2 | 51.6 | 356.8 | 50.1 | 34.2 |
| WiSE-FT | 82.4 | 112.8 | 84.2 | 46.6 | 57.7 | 373.6 | 55.1 | 40.7 |
| ImpReg | 81.9 | 108.2 | 79.7 | 46.7 | 53.0 | 376.3 | 49.9 | 34.5 |
| **PEFT** | | | | | | | | |
| Naïve | 88.2 | 17.9 | 90.7 | 59.1 | 80.1 | 38.7 | 76.7 | 67.2 |
| L2-SP | 88.3 | 19.0 | 90.5 | 58.6 | 80.3 | 39.2 | 76.4 | 67.3 |
| FTP | **88.6** | 26.1 | **92.0** | **59.7** | **80.3** | **10.8** | **80.5** | **68.0** |
| WiSE-FT | 88.0 | 22.5 | 90.5 | 59.3 | 80.3 | 39.3 | 77.0 | 67.3 |
| ImpReg | 86.2 | **16.5** | 89.8 | 56.3 | 77.4 | 27.0 | 77.2 | 64.4 |

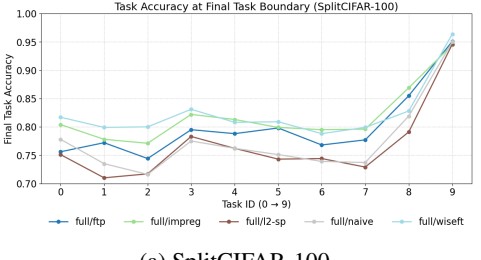

(a) SplitCIFAR-100.

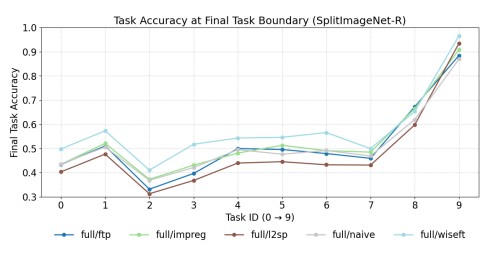

(b) SplitImageNet-R

Figure 3: Final task accuracies across 10 tasks for CLIP-ResNet-50 under FullFT. (a) We compare the final task (LP) accuracies $A_{T-1,i}$ for $i = 0, \ldots, 9$ when CLIP-ResNet-50 is fully fine-tuned. Naïve fine-tuning and L2-SP exhibit representational forgetting, while FTP, ImpReg, and WiSE-FT reduce forgetting and are the best models. (b) We observed a similar trend in SplitImageNet-R where WiSE-FT and ImpReg are the best performing methods.

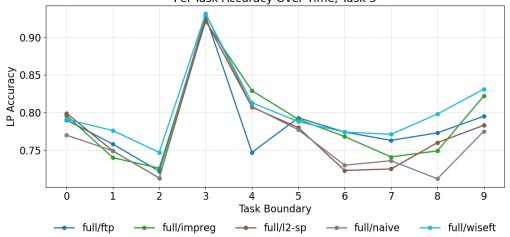

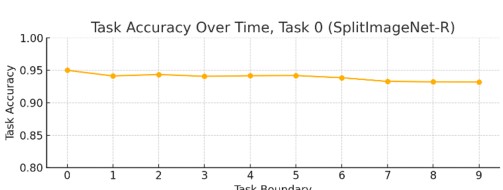

Figure 4: 10-Task SplitCIFAR-100. Forgetting observed for task $T_3$ after the $T_4$ boundary, indicating rapid degradation.

Figure 5: SplitImageNet-R. Minimal forgetting for task $T_0$ across the sequence under Naïve PEFT fine-tuning.

## 4.2 RESULTS FOR FEATURE-PRESERVING METHODS UNDER PEFT STRATEGY

In the previous section, we showed that FullFT, even with feature-preserving methods, leads to representational forgetting. Here we investigate preventing representational forgetting under the PEFT strategy, where the ResNet-50 feature extractor is frozen and only the attention pooling block is updated. This design is motivated by prior work (Smith et al., 2023a) and prompting approaches such as L2P (Wang et al., 2022a) and DualPrompt (Wang et al., 2022b), which have shown reduced forgetting by updating a small subset of parameters. Specifically, we apply L2-SP, FTP, WiSE-FT, and ImpReg to the trainable parameters of the attention pooling block and evaluate performance.

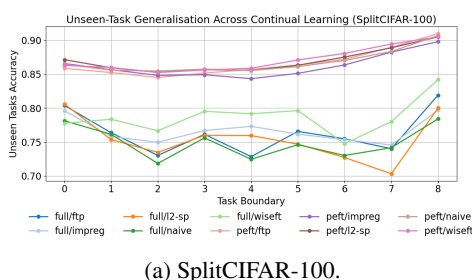 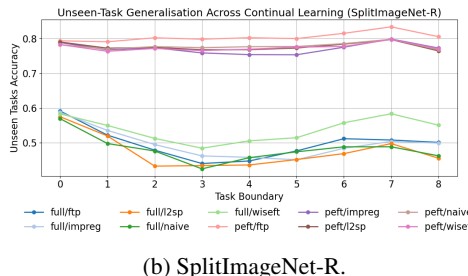

(a) SplitCIFAR-100.             (b) SplitImageNet-R.

Figure 6: UTA across task boundaries, showing PEFT consistently outperforming FullFT.

Our results in Table 1 (PEFT) show that PEFT leads to a significant reduction in representational forgetting (RF) and improved generalisation to unseen tasks (AvgUTA), resulting in higher final task accuracy (FTA) across both SplitCIFAR-100 and SplitImageNet-R compared to their FullFT counterparts. Overall, FTP emerges as the most effective feature-preserving methods when applied to the attention-based pooling block across both benchmarks, with all methods performing reasonably well. Interestingly, we found that the performance gain from adding the feature-preserving constraint is almost the same when compared to Naïve fine-tuning. This suggests that the attention pooling block may be inherently resistant to representational forgetting when it is being fine-tuned.

To further validate this, we performed an extensional experiment – we train the first task $T_0$ using Naïve FullFT, then we freeze the ResNet-50 Feature Extractor and perform PEFT for all subsequent tasks. To our surprise, the results (Fig. 5) showed that the model displayed almost zero representational forgetting for task $T_0$, even in the absence of feature-preserving constraint. This is consistent with our results in Table 1, where RF is substantially reduced in PEFT when compared to FullFT.

Although PEFT largely prevents representational forgetting and substantially improves unseen-task generalisation, it introduces a clear trade-off in plasticity. The immediate LP accuracy following each task update is lower than under FullFT, reflecting the limited capacity of the model to encode new features when only a small subset of parameters is updated. This behaviour aligns with existing literature: while freezing most of the encoder preserves stability, it constrains the extraction of discriminative information to the attention pooling mechanism, which may be insufficient for certain tasks. These observations motivate us to explore whether it is possible to enhance the plasticity of the pre-trained model under PEFT while maintaining its resistance to forgetting. In the next section, we introduce a novel strategy that addresses this by combining PEFT with feature-preserving FullFT.

## 5 PROPOSED METHOD: PEFT+CONS FOR FEATURE PRESERVING IN CONTINUAL LEARNING

We propose PEFT+Cons, a novel two-stage fine-tuning strategy, shown in Fig. 7 which combines PEFT with consolidation of task specific knowledge (**PEFT**+Knowledge-**Cons**olidation). In the first stage, it fine-tunes the model with PEFT to reduce representational forgetting and enhance generalisation. In the second stage, it uses FullFT constrained by feature-preserving techniques to consolidate task-specific knowledge. The intuition behind PEFT+Cons is the following:

- **Stage 1:** The attention pooling block is updated under PEFT, which our experiments (Sec. 4.2, Table 1, Fig. 6a–b) show leads to representations that are more robust to forgetting while maintaining strong generalisation. By freezing the feature extractor, PEFT stabilises previously learned features at the cost of reduced per task accuracy.
- **Stage 2:** FullFT is then applied with feature-preserving methods, enhancing features not covered in Stage 1 and capturing more task-relevant information. As shown in Sec. 4.1, these methods achieve high within-task accuracy, matching or exceeding Naïve fine-tuning, while remaining more robust to forgetting, making Stage 2 a *consolidation phase* that restores plasticity without destabilising the representation space.

The PEFT+Cons results are presented in Table 2 and compared with the best methods from Table 1 for the FullFT and PEFT strategies. They show that the proposed PEFT+Cons strategy, when

Table 2: Comparison of the proposed **PEFT+Cons** method with the best FullFT and PEFT methods from Table 1 on SplitCIFAR-100 and SplitImageNet-R.

| Method | SplitCIFAR-100 | | | | SplitImageNet-R | | | |
|---|---|---|---|---|---|---|---|---|
| | FTA | RF | AvgUTA | FinalGLP | FTA | RF | AvgUTA | FinalGLP |
| **Comparison methods** | | | | | | | | |
| FullFT WiSE-FT | 82.4 | 112.8 | 84.2 | 46.6 | 57.7 | 373.6 | 55.1 | 40.7 |
| PEFT FTP | 88.6 | 26.1 | 92.0 | 59.7 | 80.3 | **10.8** | **80.5** | 68.0 |
| **PEFT+Cons (Ours)** | | | | | | | | |
| Naïve | 86.0 | 66.0 | 87.1 | 51.4 | 70.7 | 197.2 | 61.9 | 54.0 |
| L2-SP | 86.0 | 70.7 | 87.3 | 52.4 | 69.8 | 206.3 | 62.4 | 53.0 |
| FTP | **91.1** | 22.8 | **92.0** | **63.7** | **82.2** | 47.7 | 77.4 | **69.2** |
| WiSE-FT | 89.3 | 43.7 | 91.4 | 59.7 | 80.0 | 96.4 | 70.6 | 65.8 |
| ImpReg | 85.7 | 73.6 | 87.1 | 52.6 | 70.4 | 198.3 | 62.6 | 53.6 |

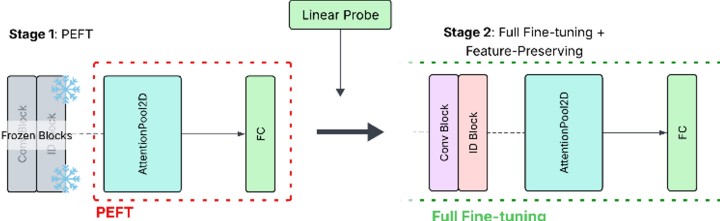

Figure 7: **PEFT+Cons:** a two-stage fine-tuning strategy designed to balance the stability/plasticity trade-offs by combining 1) PEFT and 2) FullFT with feature preserving constraints. Linear Probe shows where the task-wise representation quality is evaluated.

used with either FTP or WiSE-FT feature-preserving, leads to substantial improvements in FTA across both benchmark datasets. In particular, the combination of PEFT+Cons with FTP achieves the strongest overall performance, outperforming the other feature-preserving methods across all evaluation metrics. The PEFT+Cons methods tend to exhibit higher RF and slightly lower unseen-task generalisation (AvgUTA) compared to PEFT FTP. However, this is offset by improvements in FTA and FinalGLP, suggesting that the model is able to encode more information from each task and enhance the final representation quality.

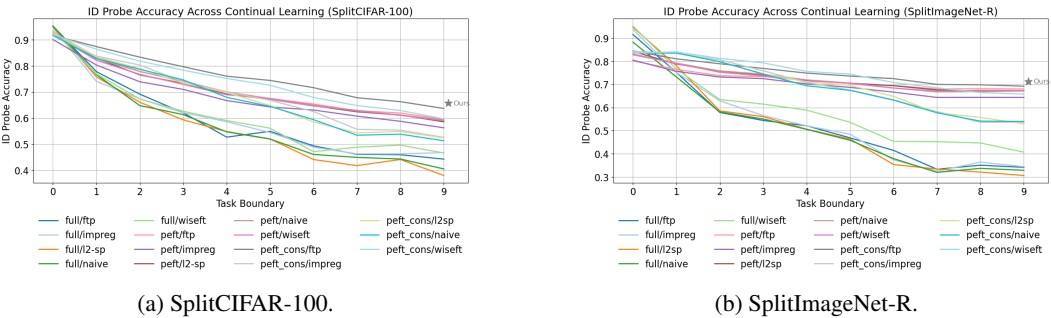

(a) SplitCIFAR-100.    (b) SplitImageNet-R.

Figure 8: ID Probe accuracy across tasks. PEFT+Cons/FTP and PEFT+Cons/WiSE-FT demonstrates high effectiveness at maintaining the stability/plasticity trade-offs.

To further verify this claim, we introduce another evaluation metric similar to GLP. Specifically, we introduce In-Distribution (ID) Probe, where we evaluate the LP accuracy restricted to in-distribution classes observed up to each task boundary. ID Probe measures both the model's plasticity and its robustness to representational forgetting over time. Using ID Probe (Fig. 8), we can see that applying PEFT+Cons with FTP achieves the highest ID Probe accuracy, particularly for early tasks, while WiSE-FT also shows strong performance.

In summary, the proposed PEFT+Cons method provides a balanced and effective strategy to incorporate feature-preserving methods into continual learning settings. By decoupling fine-tuning into two stages, it reduces the curse of representational forgetting while enabling efficient task learning.

## 6 Discussion

Our findings offer insights into the underlying mechanisms of foundation models in continual learning. We attribute the effectiveness of PEFT+Cons to two factors: (i) the resistance to forgetting in the attention pooling block under PEFT, and (ii) the feature-preserving effect of FullFT, which reduces forgetting.

To address (i): *Why does adapting only the attention pooling block resist forgetting?* We argue that this robustness comes from the fact that features near the output of the encoder are more linearly separable. Fine-tuning the pooling block therefore re-weights an embedding that is already discriminative, allowing linear probes to maintain separability across old and new tasks. This interpretation is supported by Alain & Bengio (2016), who showed that deeper layers of neural networks exhibit higher linear separability. Specifically, the AttentionPool2D layer computes a pooled representation $z = \sum_j \alpha_j h_j$, where $h_j$ are the frozen encoder outputs and $\alpha_j$ are attention weights. Because $f_\theta$ remains fixed under PEFT, $h_j$ is stable. Modifying only $\alpha_j$ changes their contributions without changing the directions of $h_j$. This preserves linear separability across tasks, whereas FullFT modifies $f_\theta$ directly and risks collapsing or rotating the feature space, leading to high representational forgetting. With this perspective, our results also help explain why FTP performs well: by projecting encoder weights back toward their original directions after each update, FTP constrains representational drift in a way that is similar to how PEFT preserves the directions of encoder features.

To address (ii): *Why does feature-preserving constraints during FullFT reduce forgetting?* We argue that its effectiveness comes from controlled plasticity through regularisation once stability has been established in Stage 1. Because the attention pooling block has already adapted to the current task, the encoder requires fewer adjustments to existing parameters. Feature-preserving constraints guide these updates to be close to the original weight space, ensuring that new information is consolidated without eroding prior representations. This is supported by our results in Table 2, where forgetting is consistently lower when FullFT is combined with feature-preserving methods, particularly FTP.

Our findings point to several directions for further investigation. First, the robustness of the attention pooling block suggests that high-level aggregation modules play a special role in stabilising representations. A deeper theoretical and empirical analysis of why such modules resist forgetting, and how their structural properties contribute to generalisation across tasks is an interesting line for future research. Second, our results suggest that because fine-tuning the encoder leads to high representational forgetting, the encoder may be where pre-trained generalisation is stored. Projection methods such as FTP help protect this ability during adaptation. Understanding how these mechanisms constrain representational drift, and whether they can be extended or combined with other regularisers, is an important direction for future work. Finally, our results highlight the value of viewing continual learning in pre-trained architectures as the interaction between high-level re-weighting components and deeper, more volatile feature extractors. Understanding this may inform the design of more robust strategies for adapting foundation models in dynamic environments.

## 7 Conclusion

In this paper, we explored the challenges of representation degradation in continual learning settings using pre-trained foundation models. We found that FullFT, even when constrained by feature-preserving methods, is insufficient for preventing significant representational forgetting, while PEFT do not provide high per-task performance. Our proposed PEFT+Cons strategy successfully balances stability and plasticity, preserving generalisable pre-trained features while allowing accurate per-task performance. Particularly, we found that the combination PEFT+Cons with FTP, produces the strongest overall performance, shows minimal representation forgetting and superior per-task accuracy. These findings underscore the importance of developing fine-tuning strategies to harness the full potential of foundation models in dynamic continual learning environments.

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
