# OpenReview forum: "Preserving Representation In Continual Learning via Feature-Preserving Fine-Tuning"
_ICLR.cc/2026/Conference — Submitted to ICLR 2026_

### Official Review · Reviewer_6xnY · 2025-10-30

**Soundness:** 1
**Presentation:** 2
**Contribution:** 1
**Rating:** 2
**Confidence:** 4

**Summary:**

This paper investigates the challenge of representation degradation in continual learning (CL) for pre-trained foundation models. The authors systematically evaluate several feature-preserving fine-tuning methods in a class-incremental learning setting using a CLIP ResNet-50 model. It proposes a two-stage strategy, PEFT+Cons, which integrates PEFT with task-specific knowledge consolidation to better balance stability and plasticity.

**Strengths:**

This paper explored the challenges of representation degradation in continual learning with pre-trained foundation models. It proposed a two-stage fine tuning strategy that combines PEFT with task-specific knowledge consolidation, and provide insights into its effectiveness in mitigating representational forgetting.

**Weaknesses:**

The innovation of this paper is limited. This paper primarily provides an empirical exploration of how combining FullFT or PEFT with different feature-preserving methods affects the maintenance of representational capacity in continual learning. The analysis of the underlying reasons is relatively empirical and lacks an investigation into the fundamental causes; the analysis is somewhat superficial and not sufficiently in-depth. The proposed method also appears to be a straightforward combination of FullFT or PEFT with various feature-preserving techniques, raising significant doubts regarding its novelty.

Furthermore, the method proposed in this paper is not compared with other approaches based on pre-trained models. The datasets and continual learning scenarios used for validation are too limited, making the experimental support insufficient. In the Related Work section, a comprehensive analysis of current continual learning methods based on pre-trained models should be provided, but the discussion of such methods in this section is not thorough.

**Questions:**

See Weaknesses.

---

> ### Author Response · Authors · 2025-11-20
>
> Dear Reviewer 6xnY
>
> We very much thank the reviewer for the thoughtful and constructive feedback. Please find our response below:
> ## Regarding the novelty of our paper
> We believe that our paper provides sufficient contribution and innovation, as we uncover architectural and representational behaviours in large pre-trained models that have not been examined in prior continual learning work. Following these observations, we proposed a novel method that follows logically from these findings.
>
> We would like to emphasise that, our novel method is not simply a recombination of FullFT and PEFT. Instead, the novelty lies in the insights that emerged from our experiments and results through our proposed metrics. While prior methods applied feature-preserving regularisation within either FullFT or PEFT, our study shows that these settings fail for different reasons. FullFT suffers from representational drift in a rehearsal-free continual learning setting despite being constraint with feature-preserving techniques, while PEFT preserves representations but at the cost of reduced plasticity to learn new tasks.
>
> Our study highlights several observations that, to our knowledge, have not been documented in prior work:
> - We show that fine-tuning only the attention-pooled CLIP encoder (PEFT) results in near-zero representational forgetting across 10 tasks (Fig. 5, Table 1), even when no feature-preserving techniques (L2-SP, FTP, WiSE-FT, ImpReg) are applied.
> - Our experiments indicate that representational forgetting arises almost entirely from updates to the feature extractor, even when feature-preserving methods are used. Restricting updates to the pooling block keeps representations stable but limits plasticity. This explains why feature-preserving FullFT alone cannot reliably prevent representational drifts and motivates the two-stage approach in PEFT+Cons
> - The contribution of PEFT+Cons lies in bringing these two observations together: Stage 1 uses PEFT to stabilise the pretrained representations, and Stage 2 applies feature-preserving FullFT to restore plasticity. We show that this sequential design produces a stability-plasticity balance that neither stage achieves on its own.
>
> We believe these empirical results and analyses offer useful insight into the representation-level behaviour of large pretrained networks in continual learning settings. We uncovered architectural behaviour of CLIP-ResNet-50 that emerges through our results in several metrics (RF, FTA, AvgUTA, FinalGLP). The design of PEFT+Cons follows logically from our empirical findings.
>
> Based on these findings, we introduce a practical and well-supported method that applies feature-preserving techniques in an empirically grounded way. In particular, our approach offers a novel strategy for incorporating feature-preserving constraints into the continual fine-tuning of large pre-trained models, when applying these techniques within FullFT or PEFT alone does not produce satisfactory accuracy.
>
> ## Regarding comparison with other approaches
> The main contribution of our work is focused on understanding the representational behaviour of large pre-trained models under continual fine-tuning. This focus on representation-level behaviour makes direct comparison to other continual learning methods such as replay-based, prompt-tuning or modular approaches less meaningful. Replay-based methods, whilst proven to be effective, maintain performance by storing and reusing past data rather than by using the pretrained representation itself. Prompt-tuning and other modular approaches introduce additional components to the model architecture rather than modifying the existing parameters. In contrast, our work studies how these properties emerge when the model is constrained to a rehearsal-free, fixed-capacity encoder.
>
> Furthermore, our study examines how feature-preserving techniques themselves behave in continual learning scenarios, and to this end we compare multiple widely used feature-preserving methods (L2-SP, FTP, WiSE-FT, ImpReg) under both FullFT and PEFT settings. This allows us to isolate and analyse their effects on the representational level, which is the primary objective of our work.

---

### Official Review · Reviewer_Avj8 · 2025-11-01

**Soundness:** 3
**Presentation:** 3
**Contribution:** 2
**Rating:** 6
**Confidence:** 3

**Summary:**

This paper investigates the representation degradation problem in continual learning of large pre-trained models, focusing on how fine-tuning strategies affect the preservation of generalizable representations. They find that while feature-preserving methods mitigate catastrophic forgetting under FullFT, representation drift still occurs. In contrast, PEFT substantially reduces forgetting but at the cost of lower task-specific accuracy. To address this trade-off, the paper proposes PEFT+Cons, a novel two-stage fine-tuning strategy that first performs PEFT to stabilise pre-trained features, then applies feature-preserving FullFT for task-specific consolidation.

**Strengths:**

1. The introduction of representation-level metrics (RF, UTA, FinalGLP) extends evaluation beyond accuracy, offering more interpretable and transferable measures for representation stability.
2. The two-stage PEFT+Cons procedure is intuitive, reproducible, and well-motivated by observed limitations of both FullFT and PEFT.
3. The authors’ interpretation of the attention pooling block as a naturally stable representation aggregator provides useful intuition that may inspire architectural research in continual learning.

**Weaknesses:**

1. The study’s conclusions are based solely on CLIP-ResNet-50 backbones and vision-only tasks. It remains uncertain whether the same representational preservation trends would hold for other backbones and other tasks.
2. While the empirical findings clearly demonstrate the stability benefits of feature-preserving fine-tuning, the paper provides no theoretical framework explaining why these methods maintain representational similarity in continual settings.

**Questions:**

1. Can the consolidation stage be scheduled adaptively based on a forgetting metric rather than fixed intervals?
2. Could combining PEFT+Cons with replay-based methods further improve the stability–plasticity trade-off?

---

> ### Author Response · Authors · 2025-11-20
>
> Dear Reviewer Avj8
>
> We sincerely appreciate the reviewer’s constructive comments. Please find our response below:
> > The study’s conclusions are based solely on CLIP-ResNet-50 backbones and vision-only tasks …
>
> We agree that evaluating additional backbones and other task modalites could further enhance our experiments. However, our primary objective is to examine the behaviour of feature-preserving methods in class-incremental learning, and using a single, well-established architecture allows us to conduct a focused and interpretable analysis. ResNet-50 serves as a suitable and commonly adopted backbone for this setting, and consistent with prior continual learning research.
>
> > Q1: Can the consolidation stage be scheduled adaptively based on a forgetting metric rather than fixed intervals?
>
> We agree that, in principle, the consolidation stage could be scheduled adaptively based on a forgetting metric rather than at fixed task boundaries. However, in our setting this is non-trivial, as computing a forgetting metric typically requires access to past-task data or stored exemplars in order to approximate the performance (or representation quality) on earlier tasks. This violates our rehearsal-free setting, where no past samples are stored and no replay buffer is maintained.
>
> If forgetting signals could be computed or approximated without access to past data, an adaptive, metric-driven scheduling, possibly with replay-based methods would indeed be a very interesting direction for future research.
>
> > Q2: Could combining PEFT+Cons with replay-based methods further improve the stability–plasticity trade-off?
>
> As addressed in Q1, this is indeed possible and an interesting line of reserach. Furthermore, using replays to reinforces past task information during either stages would help further stabilise representations while enabling more plasticity during consolidation. However, our work focuses specifically on the rehearsal-free setting in order to isolate the representational behaviour of feature-preserving techniques without relying on stored exemplars. As discussed in Section 2, replay-based methods pose practical challenges in large pre-trained models. In particular, the original pretraining data for foundation models such as CLIP is often proprietary and inaccessible to the public domain, making standard replay strategies infeasible to protect the pre-trained features when fine-tuning is applied to the feature extractor.

---

### Official Review · Reviewer_zz79 · 2025-11-01

**Soundness:** 2
**Presentation:** 3
**Contribution:** 2
**Rating:** 4
**Confidence:** 4

**Summary:**

This paper presents a framework for preserving representations in continual learning, addressing the degradation of pre-trained foundation models when fine-tuned sequentially. The authors systematically evaluate several feature-preserving fine-tuning (FPFT) techniques (L2-SP, FTP, WiseFT, and ImpReg) in class-incremental learning with CLIP-ResNet-50. They highlight that naive full fine-tuning (FullFT) causes severe representational forgetting, while parameter-efficient fine-tuning (PEFT), which only updates the attention pooling block, greatly alleviates this issue but limits adaptability to new tasks. To balance stability and plasticity, the paper introduces PEFT+Cons, a two-stage fine-tuning strategy that first performs PEFT to stabilize pre-trained features, then applies feature-preserving FullFT to consolidate task-specific knowledge. This approach enables both robust representation retention and effective adaptation. Experiments on Split CIFAR-100 and Split ImageNet-R demonstrate that PEFT+Cons with FTP achieves the best trade-off, substantially improving final task accuracy and representational robustness while maintaining strong generalization to unseen tasks. Overall, the study provides new insights into how feature-preserving regularization and modular fine-tuning interact in foundation models. It underscores the importance of designing fine-tuning strategies that protect pre-trained representations while supporting continual adaptation, marking a step toward more stable and generalizable continual learning with vision-language foundations.

**Strengths:**

1.The paper is clearly written, and its overall structure flows logically from the underlying motivation and problem formulation to the methodological design and comprehensive experimental validation.

2.The experimental design is rigorous and goes beyond conventional evaluations based solely on classification accuracy. Instead, it employs a comprehensive set of representation-level metrics—such as Representational Forgetting (RF), Unseen Task Accuracy (AvgUTA), and Global Linear Probe Accuracy (FinalGLP)—thereby providing a more nuanced and convincing assessment of the model’s effectiveness.

3.Building on comprehensive experimental analyses, the paper introduces PEFT+Cons, a novel two-stage fine-tuning strategy designed to achieve a balanced trade-off between stability and plasticity. In this approach, Stage 1 employs PEFT to preserve robust representations and maintain generalization, while Stage 2 applies a constrained FullFT to consolidate task-specific knowledge. Empirical results demonstrate that FTP with this strategy consistently outperforms other fine-tuning methods across multiple evaluation metrics.

4.The paper carefully characterizes the behaviors of both FullFT and PEFT strategies through well-designed experiments, offering valuable insights into their respective strengths and limitations. This analysis provides a clear rationale for why the proposed PEFT+Cons method achieves better performance under continual learning scenarios.

**Weaknesses:**

1.The empirical evaluation uses only two datasets (SplitCIFAR-100 and SplitImageNet-R) and a single backbone (CLIP-ResNet-50). While these are reasonable starting points, the paper does not demonstrate whether conclusions generalize to (a) other backbone families (e.g., ViTs or larger CLIP variants), (b) longer task sequences or different class granularities, and (c) warm start setting[1].

2.The work attributes PEFT’s resistance to forgetting to properties of the attention pooling block, yet does not examine alternative PEFT designs or architectural modifications that might further improve plasticity without FullFT (e.g., small adapter modules, layer-wise low-rank updates, or selective unfreezing). It would be beneficial for the authors to conduct additional experiments to investigate the effectiveness of PEFT+Cons with other modules which are commonly used in peft PTM methods[2,3].

3.The paper reports that the combination of FTP with PEFT+Cons yields the best overall performance, yet provides limited mechanistic explanation for why FTP consistently outperforms other feature-preserving approaches within this two-stage framework. Moreover, the performance of other methods under the PEFT+Cons strategy is generally lower on the FTA, AvgUTA, and FinalGLP metrics compared to their results under the PEFT strategy without consolidation. A deeper analysis of these results would offer valuable insights and strengthen the authors’ conclusions.

4.The experiments presented in this paper show that the FTP combined with the PEFT+Cons strategy achieves the best performance on two benchmark datasets compared with other fine-tuning methods. However, this evidence alone is not fully convincing. The applicability and generality of the proposed approach could be further strengthened by evaluating its performance with additional continual learning methods [4–7] (e.g., LwF, EWC, iCaRL, ZSCL).

[1] Elastic feature consolidation for cold start exemplar-free incremental learning.
[2] Expandable Subspace Ensemble for Pre-Trained Model-Based Class-Incremental Learning.
[3] LoRA Subtraction for Drift-Resistant Space in Exemplar-Free Continual Learning.
[4] Learning without Forgetting
[5] Overcoming catastrophic forgetting in neural networks
[6] iCaRL: Incremental Classifier and Representation Learning
[7] Preventing Zero-Shot Transfer Degradation in Continual Learning of Vision-Language Models

**Questions:**

1.Why was CLIP-ResNet50 pre-trained on ImageNet-1K chosen as the backbone? Given that CLIP models are already trained on large-scale and diverse datasets, the necessity of additional pretraining on ImageNet-1K is unclear. Furthermore, evaluating the proposed method with different backbone architectures would further validate its generality and strengthen the evidence for its effectiveness.

2.All experiments appear to have been conducted using a single random seed, raising concerns that the chosen hyperparameter settings may be overfitted to this specific configuration. Evaluating the method across multiple random seeds would provide a more reliable assessment of its robustness and generalization.

3.As shown in Table 1, the RF of the naive method combined with PEFT strategy is lower than that of L2-SP, FTP, and WiSE-FT on the benchmark datasets. What accounts for the reduced forgetting observed in the Naïve method? Moreover, the final GLP of the naive method is comparable to that of other approaches while requiring significantly less computational cost. What factors contribute to these results?

---

> ### Author Response · Authors · 2025-11-20
>
> Dear Reviewer zz79
>
> We are grateful to the reviewer for providing us with a thoughtful and constructive feedback. Please find our response below:
>
> > Q1a:  Why was CLIP-ResNet50 pre-trained on ImageNet-1K chosen as the backbone? Given that CLIP models are already trained on large-scale and diverse datasets, the necessity of additional pretraining on ImageNet-1K is unclear
>
> To clarify, we used the publicly released CLIP-ResNet-50 checkpoint, whose visual encoder is based on a ResNet-50 that was originally pretrained on ImageNet-1K as part of CLIP. We did not perform any extra pretraining. In the camera-ready version, we will clarify this with more appropriate wording.
>
> >Q1b: Furthermore, evaluating the proposed method with different backbone architectures would further validate its generality and strengthen the evidence for its effectiveness.
>
> We agree that evaluating additional backbones could further strengthen the study. However, our work focuses specifically on analysing the effectiveness of feature-preserving methods under class-incremental learning. We believe that selecting ResNet-50 as a designated backbone for our study is an appropriate and widely adopted choice for this purpose, and many continual-learning works referenced in our paper similarly restrict their analysis to a designated backbone to ensure controlled and interpretable comparisons.
>
> >Q2: All experiments appear to have been conducted using a single random seed … Evaluating the method across multiple random seeds would provide a more reliable assessment of its robustness and generalization.
>
> This is correct - while the current experimental results uses a single random seed, we believe this is sufficient for the purpose of our study. This choice was made to keep the experimental scope focused while performing extensive analysis on the representation-level. Our experiments do not tune the hyperparameters for each method. Instead, we use the default hyperparameters from the original implementations of L2-SP, FTP, WiSE-FT, and ImpReg, and apply these consistently across FullFT, PEFT, and PEFT+Cons to evaluate each method in isolation. The observations made in our paper, such as the near-zero representational forgetting under PEFT, representational degradation in FullFT and increase in plasticity of PEFT+Cons are highly consistent across tasks and datasets, making it unlikely that multi-seed experiments would deviate from the reported results.
>
> Nonetheless, we agree that evaluating across multiple seeds would improve the robustness and reliability of our conclusions. Due to computational constraints, we were unable to include a full multi-seed analysis, but we will do this in future extensions of the work.
>
> >Q3 As shown in Table 1, the RF of the naive method combined with PEFT strategy is lower than that of L2-SP, FTP, and WiSE-FT on the benchmark datasets … What factors contribute to these results?
>
> We attribute the robustness in RF in the naïve method to the architectural behaviour of CLIP-ResNet-50 under PEFT, rather than to the absence or presence of a specific feature-preserving technique. Specifically, our analysis indicates that the AttentionPool2D block weights tends to be more discriminative and stable. Because the encoder is frozen, the block does not modify feature directions but only adjusts attention weights over fixed representations. This results in low RF even without adding any feature-preserving constraints. As a result, methods like L2-SP, FTP, and WiSE-FT offer limited additional benefit in FinalGLP accuracy when applied only to the small parameter subset in the AttentionPool2D block.
>
> This observation directly motivates the development of the new approach PEFT+Cons. While PEFT alone preserves stability, it also introduces reduced plasticity, as the frozen encoder limits the model’s ability to extract new task-specific features. PEFT+Cons leverages feature-preserving techniques by first encoding a stable representational base in the AttentionPool2D block, and the subsequent feature-preserving FullFT stage carefully restores plasticity while constraining updates to remain close to the pretrained feature space. The feature-preserving methods become effective in Stage 2 precisely because they operate after stability has been secured with PEFT in Stage 1.

---

### Official Review · Reviewer_AvJk · 2025-11-01

**Soundness:** 2
**Presentation:** 2
**Contribution:** 2
**Rating:** 4
**Confidence:** 3

**Summary:**

This paper investigates the problem of representation degradation in continual learning with pre-trained foundation models. The authors systematically evaluate four recent feature-preserving fine-tuning approaches (L2-SP, FTP, WiseFT, and ImpReg) under class-incremental continual learning using a CLIP-ResNet-50 backbone on Split CIFAR-100 and ImageNet-R. They compare full fine-tuning (FullFT) and parameter-efficient fine-tuning (PEFT) and introduce a two-stage method (PEFT+Cons) that combines PEFT with feature-preserving consolidation. Results indicate that PEFT+Cons, particularly when paired with FTP, provides improved balance between stability and plasticity, reducing representational forgetting without substantially constraining task-specific adaptation.

**Strengths:**

1. The systematic comparison of FullFT and PEFT strategies reveals nuanced insights into when and why representational forgetting occurs, using well-chosen metrics.
2. Demonstrates up to over FTA points over the next-best method
3. Figures provide a clear illustration of the improvement induced by the proposed approach.

**Weaknesses:**

Limited Novelty in Core Mechanisms: The PEFT+Cons procedure, while well-executed and carefully evaluated, essentially combines PEFT and existing feature-preserving FullFT approaches in a sequential pipeline. It does not introduce fundamentally new algorithms or theoretical insights into representation preservation. The novelty largely resides in the two-stage orchestration, not in a new method or principle.

**Questions:**

Can the authors justify the novelty of the proposed approach over the simple recombination of existing approaches from the same domain?

---

> ### Author Response · Authors · 2025-11-20
>
> Dear Reviewer AvJk
>
> We thank the reviewer for their considerate and valuable to our work. Please find our response below:
>
> > Can the authors justify the novelty of the proposed approach over the simple recombination of existing approaches from the same domain?
>
> The primary contribution of our work lies in the carefully designed experimental analysis, which provides empirically grounded insights into how feature-preserving techniques can be applied effectively and efficiently in continual fine-tuning settings. Specifically, we first conduct a series of experiments that reveal several new underlying properties of how large pre-trained models behave during fine-tuning at both the architectural and representation levels. Based on these observations, we proposed a novel method that makes use of feature-preserving techniques in a two-stage manner - a use of these techniques that, to our knowledge, has not been explored in prior work. Our approach addresses the limitations of applying feature-preserving techniques within FullFT or PEFT strategies alone and introduces a practical strategy for stabilising and adapting large pre-trained models under continual fine-tuning.
>
> Our study reveals several findings that, to our knowledge, have not been reported in prior work:
>
> - We show that fine-tuning only the attention-pooled encoder (PEFT) achieves near-zero representational forgetting across 10 tasks (Fig. 5, Table 1), even for naïve PEFT without any feature-preserving techniques (L2-SP, FTP, WiSE-FT, ImpReg).
> - Our experiment revealed that forgetting comes almost entirely from updates to the feature extractor, even when feature-preserving methods are applied. Updating only the pooling block keeps representations stable, though at the cost of reduced plasticity. This clarifies why feature-preserving FullFT alone cannot prevent drift and motivates our two-stage procedure.
> - The novelty of PEFT+Cons lies in the integration of our two findings: PEFT is first used to stabilise pretrained representations (Stage 1), after which feature-preserving FullFT is applied to recover plasticity (Stage 2). We show that this sequential design achieves a stability-plasticity balance that neither component achieves alone.
>
> We believe these empirical findings and analyses provide meaningful insight into the representation-level behaviour of large pretrained networks under continual learning settings. Building on these observations, we proposed an effective and well-supported novel method that leverages feature-preserving techniques in a novel and empirically grounded way, achieving stronger representation retention and higher final accuracy than existing approaches.

---

### Meta-Review · Area_Chair_TazZ · 2026-01-01

**Summary:**

The primary concerns raised by the reviewers focused on the limited novelty of the proposed PEFT+Cons strategy, which is considered as a straightforward combination of existing parameter-efficient and feature-preserving fine-tuning techniques. Concerns of reviewers include experiment with only a single backbone (CLIP-ResNet50), a single random seed, and two datasets, which raises questions about the robustness and generalizability of the proposed method. Additionally, there are concerns regarding the lack of theoretical analysis to explain the experimental results and the absence of comparison with a broader range of continual learning baselines, such as prompt tuning or replay-based methods.

**Reviewer Concerns:**

The authors tried to address the novelty concern by clarifying that their contribution stems from empirical insights into the distinct roles of PEFT (stability) and FullFT (plasticity), which motivates the two-stage design. However, this implies the contribution of this work lies primarily on engineering implementation rather than algorithm design. The concerns regarding the limited experimental scope remain outstanding, as the authors acknowledged the lack of multi-seed and multi-backbone experiments but did not provide new results to validate robustness. Similarly, the lack of theoretical analysis and broader baseline comparison remains unresolved.

**Reviewer Scores:**

Reviewer AvJk (Score 4) and Reviewer 6xnY (Score 2) would likely maintain their scores, as the rebuttal confirms the method is a simple combination of existing techniques. Reviewer zz79 (Score 4) is unlikely to raise their scores because the specific requests for experiments with different backbones and multiple seeds to prove generalization are not met. Reviewer Avj8 (Score 6) might maintain a borderline assessment.

---

### Decision · Program_Chairs · 2026-01-26

Reject